# Evaluation of the Heat Shock Protein 90 Inhibitor Ganetespib as a Sensitizer to Hyperthermia-Based Cancer Treatments

**DOI:** 10.3390/cancers14215250

**Published:** 2022-10-26

**Authors:** Enzo M. Scutigliani, Yongxin Liang, Marloes IJff, Hans Rodermond, Xionge Mei, Miriam P. Korver, Vaneesha S. Orie, Ron A. Hoebe, Daisy I. Picavet, Arlene Oei, Roland Kanaar, Przemek M. Krawczyk

**Affiliations:** 1Department of Medical Biology, Amsterdam University Medical Centers, Cancer Center Amsterdam, University of Amsterdam, Meibergdreef 9, 1105 AZ Amsterdam, The Netherlands; 2Department of Molecular Genetics, Oncode Institute, Erasmus MC Cancer Institute, Erasmus University Medical Center, Doctor Molewaterplein 40, 3015 GD Rotterdam, The Netherlands; 3Laboratory for Experimental Oncology and Radiobiology (LEXOR), Center for Experimental and Molecular Medicine (CEMM), Amsterdam University Medical Centers, Cancer Center Amsterdam, University of Amsterdam, P.O. Box 22700, 1100 DE Amsterdam, The Netherlands; 4Department of Radiation Oncology, Amsterdam University Medical Centers, Cancer Center Amsterdam, University of Amsterdam, P.O. Box 22700, 1100 DE Amsterdam, The Netherlands

**Keywords:** hyperthermia, heat stress response, heat shock protein 90, ganetespib

## Abstract

**Simple Summary:**

Hyperthermia boosts the effects of radio- and chemotherapy regimens, but its clinical potential is hindered by the ability of (cancer) cells to activate a protective mechanism known as the heat stress response. Strategies that inhibit its activation or functions have the potential, therefore, to improve the overall efficacy of hyperthermia-based treatments. In this study, we evaluated the efficacy of the HSP90 inhibitor ganetespib in promoting the effects of radiotherapy or cisplatin combined with hyperthermia in vitro and in a cervix cancer mouse model.

**Abstract:**

Hyperthermia is being used as a radio- and chemotherapy sensitizer for a growing range of tumor subtypes in the clinic. Its potential is limited, however, by the ability of cancer cells to activate a protective mechanism known as the heat stress response (HSR). The HSR is marked by the rapid overexpression of molecular chaperones, and recent advances in drug development make their inhibition an attractive option to improve the efficacy of hyperthermia-based therapies. Our previous in vitro work showed that a single, short co-treatment with a HSR (HSP90) inhibitor ganetespib prolongs and potentiates the effects of hyperthermia on DNA repair, enhances hyperthermic sensitization to radio- and chemotherapeutic agents, and reduces thermotolerance. In the current study, we first validated these results using an extended panel of cell lines and more robust methodology. Next, we examined the effects of hyperthermia and ganetespib on global proteome changes. Finally, we evaluated the potential of ganetespib to boost the efficacy of thermo-chemotherapy and thermo-radiotherapy in a xenograft murine model of cervix cancer. Our results revealed new insights into the effects of HSR inhibition on cellular responses to heat and show that ganetespib could be employed to increase the efficacy of hyperthermia when combined with radiation.

## 1. Introduction

Cancer treatments using hyperthermia (i.e., exposure to temperatures of 41–42 °C for 1 h) as a radio- and chemotherapy sensitizer are gaining popularity for an expanding variety of tumor subtypes, including cervix, bladder, breast, head, and pharynx [1,2,3]. The enhanced effects of radio- and chemotherapy are linked to numerous biological effects of hyperthermia at the macroscopic and microscopic scale [4,5], including improved tissue perfusion [6], the inhibition of DNA repair pathways [7,8] and immune activation [9].

The clinical potential of hyperthermia is attenuated by the ability of (cancer) cells to activate a protective mechanism known as the heat stress response (HSR) [10,11]. First, activation of the HSR confers thermotolerance—transient resistance to successive heat exposures—which has been found to reduce thermo-sensitization [12,13], and is among the major reasons for generally limiting the frequency of therapeutic hyperthermia sessions to one or two per week [14]. Second, to effectively sensitize the tumor to chemo- and radiotherapy, it has to be exposed to a sufficiently high thermal dose, i.e., the product of temperature and treatment duration [15,16]. It remains challenging, however, to administer the required thermal dose uniformly across the entire tumor volume [17,18]. Consequently, the activation of the HSR allows cancer cells to recover from the often-suboptimal heating, preventing the full exploitation of the sensitizing effects. Third, due to the activation of the HSR, sensitization by hyperthermia is temporary, producing a short therapeutic window. Indeed, the time interval between the application of hyperthermia and radiotherapy is negatively correlated with overall survival of women with locally advanced cervical cancer [19], and has a larger influence on treatment efficacy than the order of application [20,21]. The HSR thus appears to interfere with therapies based on hyperthermia, and blocking its activation or functions has the potential to improve the overall efficacy [5].

The HSR results in the upregulation of several functionally different classes of proteins, originating from 50–200 genes involved in protein homeostasis, metabolism, and DNA repair, among others [10,22,23,24,25,26,27,28], while repressing more than 1000 genes that regulate cell growth, transcription, and translation [29,30,31,32]. Of these classes, the molecular chaperones are the most rapidly and highly induced proteins to facilitate rapid recovery of proteostasis by ensuring refolding or degradation of dysfunctional proteins [10,33]. Accordingly, inhibition, knockdown, or knockout of molecular chaperones, has thermo-sensitizing effects in various in vitro and in vivo tumor models of the pancreas, breast, cervix, skin, prostate, ovary, head, pharynx, and colon [34,35,36,37,38,39,40,41,42,43,44,45]. The emerging role of molecular chaperones in cancer progression [46,47] has, on the other hand, intensified the development of drugs that target specific members (e.g., HSP90 and HSP70) or related transcription factors (e.g., HSF1) [48]. The development of inhibitors targeting HSP90 has yielded highly potent compounds with a favorable safety profile, such as ganetespib, NVP-AUY992, and onalespib [49,50,51,52,53,54,55]. Inhibition of molecular chaperones, especially HSP90, may thus be an attractive approach to improve the efficacy of hyperthermia-based therapies.

In the previous proof-of-concept study [56], we demonstrated that pharmacological inhibition of HSP90 by ganetespib promotes the effects of hyperthermia-based treatments in cervix cancer cells in vitro. Here, we validated these initial findings in multiple cell lines of different tumor origins and extended the analysis to changes in the cell proteome, using an optimized treatment protocol. Finally, we evaluated the efficacy of HSP90 inhibition in promoting the effects of radiotherapy or cisplatin combined with hyperthermia in a cervix cancer mouse model.

## 2. Materials and Methods

### 2.1. Cell Lines and Culturing Conditions

The SiHa, HeLa, MCF7, and FaDu cells, transduced with the fluorescent ubiquitination-based cell cycle indicator (Fucci2a) reporter [57,58], were maintained in EMEM (Gibco, Waltham, MA, USA). The T24, RT112 and A375 cancer cells were cultivated in DMEM (Gibco). The Sk-Mel-88 and T47D cells were maintained in RPMI-1640 (Gibco). Media were supplemented with 10% fetal bovine serum (Gibco), 100 U/mL penicillin (Gibco), 100 U/mL streptomycin (Gibco), and 2 mM L-glutamine (Gibco) unless stated otherwise. Cells were cultured at 37 °C in 5% CO_2_. The original cell lines were purchased from ATCC.

### 2.2. Hyperthermia, Radiotherapy, and Drug Treatments

Hyperthermia treatment (i.e., 42 °C for 1 h) was performed by submerging cell culture vessels in a calibrated water bath or by placing them in an incubator in 5% CO_2_. Incubation was extended by 5 min to accommodate for the time needed to reach the target temperature. Radiotherapy was performed 15 min before the start of the hyperthermia treatment by exposing cells to a ^137^Cs γ-ray source at a dose rate of 0.5 Gy/min. Unless specified otherwise, ganetespib (Synta Pharmaceuticals, West Conshohocken, PA, USA) was added 30 min before hyperthermia treatment and removed 24 h afterwards. Cisplatin (TEVA) was added 30 min before hyperthermia treatment and removed immediately afterwards.

### 2.3. Cell Viability Assays

Cells were seeded in flat-bottom 24-well plates (Greiner, Frickenhausen, Germany) 24 h prior to treatments. Five days after treatment, cells were incubated with DMEM containing 10% PrestoBlue™ Cell Viability Reagent (Invitrogen, Waltham, MA, USA) for four hours, and fluorescence was measured with a CLARIOstar^®^ Plus microplate reader (BMG Labtech, Ortenberg, Germany) using excitation/emission wavelengths of 560 ± 15/590 ± 20 nm. Blank measurements were included for background correction.

### 2.4. Clonogenic Survival Assays

Clonogenic survival assays were based on the “plating-before-treatment method” described elsewhere [59]. Briefly, cells were seeded 4–18 h before hyperthermia in flat-bottom 6-well plates (Greiner). After treatment, cells were incubated for the time that was required to allow colony formation (minimum of 50 cells) and stained with PBS containing 0.05% crystal violet (Sigma, Saint Louis, MO, USA) and 1% glutaraldehyde (Sigma).

### 2.5. Western Blots

Cells were harvested by scraping in ice-cold PBS and lysed in RIPA buffer for one hour on ice. Insoluble fractions were removed from the lysate by centrifugation (14,000 rpm, 10 min, 4 °C). Protein concentrations were determined by a Bradford or Lowry assay. Per sample, 25 μg of protein was denatured (98 °C, 10 min) in the presence of a loading buffer and β-mercaptoethanol and loaded onto an SDS-PAGE gel. Gel electrophoresis was performed by stacking the lysates at 65 V for 15 min, followed by a separation phase at 120 V for two hours. Afterwards, proteins were transferred to a nitrocellulose membrane by using the Trans-Blot Turbo Transfer System (Bio-Rad, Hercules, CA, USA). Membranes were blocked by incubation with 5% milk for one hour. Immunostaining for HSP70i and β-actin was performed overnight at 4 °C with a 1:1000 dilution of monoclonal mouse IgG and polyclonal goat IgG, respectively (sc-66048; sc-1616; Santa Cruz Biotechnology, Dallas, TX, USA). Next, membranes were incubated with a 1:15000 dilution of 680 and 800 nm infrared dyes (Li-Cor, Lincoln, NE, USA) for 30 min. All antibodies were diluted in Tris-buffered saline. Blots were visualized with an Odyssey Scanner (Li-Cor) and quantified using the built-in gel analysis function in the software platform Fiji [60].

### 2.6. Apoptosis Assays

Cells were seeded in flat-bottom 24-well plates 24 h prior to the start of treatments. Next, 24 h after treatment, the medium was replaced by supplemented FluoroBrite DMEM containing 1 μg/mL Annexin V (Adipogen, San Diego, CA, USA) and 200 nM YOYO3 (Thermo Fisher, Waltham, MA, USA). Plate lids were replaced with Breathe-Easy^®^ sealing membrane (Thermo Fisher) to minimize evaporation. Imaging and quantification were done using the IncuCyte S3 imaging platform (Sartorius, Göttingen, Germany). Phase contrast images were used to quantify cell surface area as a readout for growth. To estimate apoptosis (indicated as “apoptotic material”), the integrated fluorescence intensity of Annexin V^+^/YOYO3^+^ regions was normalized to the cell surface area.

### 2.7. Analysis of Cell Cycle Progression by Flow Cytometry and Live Cell Imaging

For flow cytometry, the SiHa and HeLa cells were seeded in 10 cm culture dishes (Greiner) 24 h before hyperthermia treatment. The cells were harvested by trypsinization, fixed with 70% ice-cold ethanol, and stored at 4 °C. One hour before flow cytometry, cells were resuspended in PBS containing 25 μg/mL RNAseI (Thermo Fisher) and 40 μg/mL propidium iodide (Thermo Fisher). Single cells were gated and analyzed using a FACSCanto™ (BD Biosciences, Franklin Lakes, NJ, USA) equipped with appropriate laser and filter sets. The PI intensity was used to estimate the cell cycle phase distribution using the built-in Watson algorithm at default settings in the software FlowJo (version 10). For live cell imaging, SiHa and HeLa cells transduced with Fucci2a were seeded in flat-bottom 24-well plates 24 h before treatment. Afterwards, mVenus and mCherry fluorescence were captured daily using a Leica DM8i wide-field fluorescence microscope equipped with appropriate filter sets, a 10×/0.25 N Plan Achromat objective (Leica, Wetzlar, Germany) and a Hamamatsu C114400 camera. The culturing medium was replaced with fully supplemented FluoroBrite DMEM (Gibco) before each round of imaging for an improved signal-to-noise ratio and to remove cell debris. Image quantification was performed with MATLAB (MathWorks, Natick, MA, USA). To segment and identify the nuclei, images from the red and green channels were first background corrected by a rolling ball filter and the noise was removed by a median filter. Both channels were merged, and masks of the nuclei were generated by performing an adaptive thresholding using the “adaptthresh” Matlab function followed by a watershed separation, and small masks were removed. The nuclei masks, together with the background-corrected red and green channel images, were used to quantify red and green fluorescence intensity per cell. The cells were subsequently classified as red, green, or both, by fitting the data into three clusters with a Gaussian mixture distribution model using the Matlab “fitgmdist” and “cluster” function.

### 2.8. Stable Isotope Labeling by Amino Acids in Cell Culture

The DMEM media for SILAC (Thermo Fisher) was supplemented with 10% dialyzed serum (Invitrogen), 1% Penicillin-Streptomycin (Gibco), 2% Ultraglutamine (Lonza, Basel, Switzerland), and 1% non-essential amino acids (Silantes, Munich, Germany). The SILAC medium was supplemented with L-lysine-D_4_ (K4), ^13^C_6_ L-Lysine (K6), ^13^C_6_ L-Arginine (R6), ^13^C_6_ ^15^N_4_ L-Arginine (R10), or unlabeled counterparts (Silantes) to establish light (K_0_R_0_), medium (K_4_R_6_) and heavy (K_6_R_10_) labeling. The HeLa cells were cultured in SILAC media for seven passages over the course of two weeks and seeded 24 h before treatment. Cells were harvested by scraping, and the pellet was frozen in liquid nitrogen before being processed for mass spectrometry. Two independent experiments were performed, and label-swap controls were included to reduce false-positive hits [61].

### 2.9. Mass Spectrometry

Cells were lysed in 1 mL of 50 mM Tris HCl containing 0.5% sodium deoxycholate (pH 8.2) and MS-SAFE™ protease and phosphatase inhibitor using a Bioruptor ultrasonicator (Diagenode, Liège, Belgium). Protein concentrations were measured using the BCA assay (Thermo Fisher). Samples were reduced with 5 mM DTT and cysteine residues were alkylated with 10 mM iodoacetamide. Proteins were extracted by acetone precipitation at −20 °C overnight, followed by centrifugation at 8000× *g* for 10 min at 4 °C. After the removal of acetone, the dried protein pellet was dissolved in 1 mL of 50 mM Tris HCl containing 0.5% sodium deoxycholate (pH 8.2), and proteins were digested with trypsin overnight at 30 °C (1:100 enzyme-to-protein ratio). Digests were acidified with 50 μL 10% formic acid (FA) and centrifuged at 8000× *g* for 10 min at 4 °C to remove the precipitated SDC. The supernatant was transferred to a new centrifuge tube. The digests were purified with C18 solid phase extraction (Sep-Pak, Waters), lyophilized, and stored at −20 °C. Next, proteolytic peptides were fractionated using high pH reverse-phase chromatography. Mass spectra of proteolytic peptides were acquired on an Orbitrap Tribrid Lumos mass spectrometer (Thermo Fisher) coupled to an EASY-nLC 1200 system (Thermo Fisher) operating in positive mode (Tune version 3.3). Peptide mixtures were trapped on a 2 cm × 100 μm Pepmap C18 column (Thermo Fisher, #164564) and separated on an in-house packed 50 cm × 75 μm capillary column with 1.9 μm Reprosil-Pur C18 beads (Dr. Maisch, Ammerbuch, Germany) at a flow rate of 250 nl/min on an EASY-nLC 1200 (Thermo Fisher), using a linear gradient of 0–32% acetonitrile (in 0.1% formic acid) during 60 or 90 min. The eluate was directly sprayed into the mass spectrometer by means of electrospray ionization. For global data dependent acquisition proteomics, full MS1 scans were recorded in the range of 375–1400 *m*/*z* at 120,000 resolution. Fragmentation of peptides with charges 2–5 was performed using higher collisional dissociation.

### 2.10. Proteomics Data Processing and Analysis

The raw data was processed with MaxQuant (version 1.6.14) by following the “minimal workflow for simple standard data sets” described elsewhere [62]. Contaminants, reverse proteins, and proteins only identified by site were removed in the software Perseus (version 1.6.14.0) following a published workflow [63]. Missing values were replaced from a normal distribution using a width of 1.3 and a downshift of 1.8. Log 2 transformed data was imported in R, and proteins with an altered abundance more or less than 0.5 and −0.5 were included for overrepresentation analyses by using the “clusterprofiler” package [64]. Molecular signatures were retrieved from the Molecular Signatures Database [65,66] using the “msigdbr” package. Set theory and visualization was performed using the “VennDiagram” package. Protein-protein interaction networks were generated using stringApp (version 1.5.1) [67] in Cytoscape (version 3.8.0) [68]. For protein-protein interaction networks, a confidence level of 0.9 was used, and 10 additional interactors were allowed to account for key interacting proteins that were not detected during the analysis.

### 2.11. Quantification of DNA Double-Strand Breaks and Micronuclei In Vitro

For γH2AX foci analysis, cells were seeded on coverslips 24 h before the start of hyperthermia treatment. After treatment, cells were washed with PBS, fixed with 2% paraformaldehyde (Thermo Fisher) in PBS, and permeabilized with PBS containing 0.1% Triton X-100 (Sigma) and 1% FBS. Immunostaining of γH2AX was performed in permeabilization buffer by a 1:100 incubation with mouse anti-phosphorylated-H2AX ser-139 (EMD Millipore, Burlington, VT, USA) for 1 h, washing with PBS, and a 30 min 1:150 incubation with anti-mouse Cy3 (Jackson, West Grove, PA, USA). Samples were counterstained by incubation with Hoechst 33342 at 5 μg/mL for 10 min and mounted on glass slides using Vectashield containing DAPI (Vectorlabs, Newark, NJ, USA). The Z-stack images were captured using a Leica DM6i wide-field fluorescence microscope equipped with appropriate filter sets, a 40×/1.25/0.75 Plan Achromat oil objective (Leica) and a Leica DFC9000 GT camera. Images were deconvolved using Huygens deconvolution software (Scientific Volume Imaging), and maximum intensity plots were used to segment nuclei and γH2AX using an in-house MATLAB script that utilizes auto-thresholding and watershed separation. For micronuclei detection and quantification, cells were seeded in flat-bottom 96-well plates (Greiner), treated as indicated earlier, and fixed at 48 h after treatment. After staining with Hoechst 33342, images were captured on a Leica THUNDER equipped with a 10×/0.32 HC PL FLuotar objective (Leica) and a Leica DFC9000 GT camera. A StarDist-based algorithm [69] was trained to detect micronuclei. Cellpose [70] was used for nuclei segmentation. The percentage of cells harboring at least one micronucleus was quantified for 10 fields of view, totaling up to at least *n* = 2000 cells per condition.

### 2.12. In Vivo Survival Studies

After an acclimation period of one week, eight week old female Hsd:Athymic Nude-Foxn1^nu^ mice (Envigo) were subcutaneously injected in the right hind leg with 50 μL of PBS containing 1.1 million SiHa cells derived from an exponentially growing culture. Once the tumor reached a size of 50 mm^3^, mice were randomly assigned to experimental groups and treated every 4th day, five times in total. Cisplatin was dissolved in PBS and injected intraperitoneally two hours before hyperthermia treatment at a dose of 2 mg/kg. Ganetespib was dissolved in a 10/18 DRD solution (10% DMSO, 18% Cremophor RH 40 (Sigma), 3.6% dextrose (Sigma) and 68.4% water) and injected intraperitoneally one hour before hyperthermia treatment at a dose of 50 mg/kg. Hyperthermia treatment (42 °C for one hour) was applied by submerging the right hind leg in a thermostatically controlled water bath during anesthesia with 2% isoflurane. Incubation was extended by 10 min to accommodate for the time needed to reach the target temperature. Radiotherapy was performed within 10 min after hyperthermia by exposing the right hind leg to a dose of 3 Gy with an X-strahl γ-ray source at a dose rate of 3 Gy/min. Tumor volume and animal weight were monitored every other day. Mice were sacrificed when the tumor reached a size larger than 1000 mm^3^ or upon meeting humane endpoints. No blinding method was used during the study.

### 2.13. Ex Vivo Quantification of 53BP1 Foci in Tumor Biopsies

Female Hsd:Athymic Nude-Foxn1^nu^ mice were inoculated as described previously. Once the tumors reached a size of 100–200 mm^3^, mice were treated once and sacrificed at different timepoints. Per condition, three tumors were collected and fixed in formalin for at least 24 h at room temperature and embedded in paraffin overnight. Sections of 4 μm were subsequently deparaffinized in xylene, hydrated by incubation in decreasing concentrations of ethanol (100%, 95%, 80%, 70%, 50%), and incubated with target antigen retrieval solution (pH 9.0, DAKO) in a microwave (720 Watts, 19 min). Slices were subsequently washed twice with PBS containing 0.5% Triton X-100 and incubated in a blocking buffer (2% BSA, 0.1% Triton X-100 in PBS) for one hour. Immunostaining of 53BP1 was performed in blocking buffer by a 1:1000 incubation with mouse anti-53BP1 (Sigma) for 90 min and a 60-min incubation with anti-mouse Alexa Fluor 488 at a 1:1000 dilution (Thermo Fisher). Samples were mounted on glass slides using Vectashield containing DAPI and stored at 4 °C until imaging. Per condition, a minimum of 600 cells was imaged with six to eight Z-stacks using a 63×/0.14 HC PL APO CS2 oil objective on a Leica SP8 confocal microscope. Nuclei and 53BP1 foci were segmented and quantified using a custom-made ImageJ script that utilizes auto-thresholding and watershed separation. The DAPI masks containing multiple nuclei were excluded from quantification.

### 2.14. Statistics

All data shown represent a minimum of three independent experiments unless stated otherwise. Significance was tested with unpaired Student’s *t*-tests or two-way ANOVA with post-hoc Tukey tests, with the exclusion of proteomics-derived protein expression differences, where a one-way ANOVA was used. Where applicable, a Benjamini–Hochberg false discovery rate correction of 0.05 was used. Bar plot error bars represent standard deviation. Box plot whiskers represent a 5–95% confidence interval.

## 3. Results and Discussion

### 3.1. Short Pre-Incubation with a Low Dose of Ganetespib Maximizes Its Hyperthermic Radio- and Chemo-Sensitization Potential

In our earlier work [56], ganetespib was added 30 min before, and removed after 60 min of incubation at 42 °C. The exposure to ganetespib in vivo, however, is likely to last much longer, as pharmacokinetic studies show that ganetespib, prodrug variants (e.g., STA-1474), and other HSP90 inhibitors (e.g., onalespib), display a serum half-life of 4–8 h [52,54,71,72,73,74]. Moreover, an elevated serum level of HSP70, a commonly used biomarker for HSP90 inhibition [75], can be detected for several days after a single-dose of ganetespib [71], suggesting that the drug is still bioactive at these timepoints. While prolonged pre-exposure to ganetespib could increase its thermo-sensitizing capacity by a more complete inhibition of HSP90, it could also trigger compensatory activation of the HSR, which is observed after inhibition of HSP90 [76,77]. It is thus important to explore and optimize the treatment protocol for achieving optimal thermo-sensitization.

To this end, we exposed cervix cancer cells to ganetespib for various durations prior to a 60 min treatment at 42 °C and evaluated the thermo-sensitizing effect of the drug (Figure 1). The results showed that a short, 30 min pre-incubation with a nanomolar range of ganetespib led to the most robust thermo-sensitization and maximally promoted the cytotoxicity of radio- and chemo-thermotherapy, whereas these effects were reduced or even eliminated after a prolonged pre-incubation (Figure 1B,C). To investigate whether the preincubation-dependent reduction in treatment efficacy could be caused by a compensatory activation of the HSR, we quantified the expression of the inducible form of HSP70, a hallmark of HSR activation. In line with previous evidence, both hyperthermia and ganetespib indeed induced the expression of HSP70 (Figure 1D,E). Importantly, exposure to ganetespib elevated HSP70 within six hours, and led to its maximal expression within 24 h, which correlates with the time at which abrogation of thermo-sensitization was observed. Together, these experiments show that a short pre-incubation with ganetespib maximizes its potential to improve therapies based on hyperthermia.

Next, we evaluated ganetespib-induced sensitization to radio- and chemo-thermotherapy in nine cell lines derived from various tumor types. We observed that 30 nM of ganetespib considerably reduced the clonogenic capacity and promoted the effects of hyperthermia in all cell lines (Figure 2E), often by several fold, as compared to hyperthermia-only- and chemo-thermotherapy regimens (Figure 2F). In support of these findings, we observed that ganetespib reduced cell growth and promoted apoptosis (Figure 2A) and led to an increased cell cycle arrest in S and G2 phase on several occasions, as can be observed by the decreased percentage of cells residing in G1 (Figure 2B–D). These results confirm that ganetespib can be utilized at concentrations that are achievable in vivo to enhance the efficacy of treatments based on hyperthermia, potentially in a range of tumor (sub)types.

### 3.2. Ganetespib Re-Sensitizes Thermotolerant Cells and Promotes the Effects of Hyperthermia-Based Treatments at Lower Thermal Doses

The acquisition of thermotolerance impedes the frequency at which hyperthermia is applied [14], underscoring the interest in treatments that counteract this phenomenon. We have previously demonstrated that the inhibition of HSP90 by ganetespib enhances the sensitivity of thermotolerant cells to hyperthermia [56]. To confirm that this also takes place after optimized pre-exposure to ganetespib, we rendered cervix cancer cells thermotolerant by heat exposure 24 h prior to hyperthermia and compared their clonogenic capacity to that of thermosensitive cells to chemo- and radio-thermotherapy (Figure 3). In line with our previous findings, we observed that ganetespib sensitized to these treatment regimens (Figure 3B).

As the administration of a sufficient thermal dose, in a uniform manner, to the entire tumor volume, frequently poses a challenge in the clinic, strategies that establish adequate sensitization of cancer cells at lower temperatures and/or shorter treatment durations could lead to improved efficacy. We evaluated, therefore, whether ganetespib could sensitize cancer cells at lower thermal doses. Cervix cancer cells treated at 40 °C in the presence of ganetespib display similar sensitivity to radiotherapy as compared to the standard target temperature of 42 °C (Figure 3C). In addition, we observed that the exposure to 41 °C for 30 min was sufficient to reach similar, or higher, sensitization to chemo- and radio-thermotherapy compared to the control (Figure 3D). In conclusion, the inhibition of HSP90 by ganetespib can potentially be used to improve the efficacy of hyperthermia-based therapies under suboptimal conditions, such as the presence of thermotolerant cells, or suboptimal delivery of the thermal dose to the tumor.

### 3.3. Combined Treatment with Ganetespib and Hyperthermia Induces Unique Proteome Changes

Inhibitors of HSP90 have pleiotropic effects as they can potentially induce degradation of hundreds of client proteins (for a continually updated list see http://www.picard.ch/downloads (accessed on 10 August 2022)). To explore the mechanisms by which ganetespib may potentiate the effects of hyperthermia-based treatments in an unbiased manner, we adopted a SILAC-based proteomics approach. After isotope labeling, HeLa cells were treated with ganetespib and/or hyperthermia and harvested at different timepoints (Figure 4A). In total, we detected ~2000 individual proteins 24 h after hyperthermia treatment, and ~4000 proteins in all other cases. The number of detected proteins was stable between replicates (not shown). The number of proteins that were differentially expressed between the various conditions, along with principal component analysis (Figure 4B–E), revealed that proteome differences mostly occurred 24 h post-treatment with the combination of ganetespib and hyperthermia. An overrepresentation analysis for Gene Ontology terms, related to biological processes (GO:BP) and molecular signature hallmarks, was then performed for each experimental condition. In line with previous reports, we observed that HSP90 inhibition affected processes related to mTOR [78], and MYC (Figure 4F,G), which was recently identified as an HSP90 client [79]. Strengthening the available evidence that pharmacological inhibition of HSP90 affects DNA repair proteins [80,81], we also found an enrichment of GO:BP terms related to DNA repair in cells treated with ganetespib alone (Figure 4F). Using hand-curated categories, however, we found that genes related to DNA repair and cell cycle progression were even more prominently enriched in cells treated with a combination of hyperthermia and ganetespib (Figure 4F and Appendix AC,D). Moreover, we found that the hallmark signature of DNA repair was uniquely enriched in the combination-therapy arm (Figure 4G). A similar approach to identify altered pathways among known HSP90 interactors yielded no conclusive results (Appendix A). The abundance of four major DNA repair factors, MRE11, PCNA, RAD21 and SMARCA1, appear to be significantly decreased by the combination of hyperthermia and ganetespib at the 24 h time-point, as compared to ganetespib alone (Figure 4E). Validation using immuno-blotting failed to confirm these results, however, (Figure 4H,I). This could be because the mass-spectrometric changes in protein abundance were relatively small (under 1.25 fold), which may not be sufficient for reliable detection using immuno-blotting. The proteome alterations are suggestive, nevertheless, of an altered activity of DNA repair pathways and cell cycle progression especially when combining ganetespib with hyperthermia.

### 3.4. Ganetespib Amplifies the Inhibition of the DNA Damage Response by Hyperthermia

To evaluate how DNA repair is affected by the optimized treatment protocol, we quantified the occurrence of yH2AX foci—a hallmark of unrepaired double-strand DNA breaks [82,83], and micronuclei—representing permanent chromosome damage in cells that accomplished mitotic division. To complement our earlier work on this subject [56], each treatment combination was assessed, and custom analysis pipelines for image segmentation allowed for the high-throughput analysis of at least 500 cells per condition. Quantification of yH2AX foci confirmed that, in most cases, cells were less able to repair double-stranded DNA breaks, specifically when exposed to chemo- and radio-thermotherapy in the presence of ganetespib (Figure 5B,C). The delayed repair did not translate into an increased occurrence of micronuclei (Figure 5D). In combination with the proteomics data, these results confirm that ganetespib promotes chemo- and radio-thermotherapy at least in part by enhancing the inhibition of DNA repair mechanisms.

### 3.5. Ganetespib Promotes the Effects of Radiotherapy Combined with Hyperthermia in a Subcutaneous Murine Model of Cervix Cancer

To validate the effects of combining ganetespib with hyperthermia in vivo, we used a cervix cancer mouse model. Briefly, 1 million SiHa cells were subcutaneously injected into the right hind leg of nude mice, which resulted in palpable tumors within approximately 14 days. Tumor volume, animal weight and survival were recorded during and after twice-weekly treatment regimens consisting of various combinations of cisplatin, radiotherapy, hyperthermia and ganetespib (Figure 6A). All treatments displayed minimal toxicity-related issues in pilot experiments, except for those where cisplatin and ganetespib were combined, and the dose of both drugs in these groups was reduced by 50%. We found that all modalities, apart from radiotherapy (*p* = 0.07), significantly improved tumor control and survival as a single agent (Figure 6B). Hyperthermia-based radio- and cisplatin therapy did not outperform the monotherapies, however, and the addition of ganetespib to hyperthermia did not result in an improved outcome. Notably, we found that the addition of ganetespib to a radio-thermotherapy regimen outperformed all other treatments in four out of six mice (Figure 6D,E), but the remaining two mice showed tumor growth and survival comparable to other treatment groups (Figure 6C). To evaluate whether this increased treatment efficacy can be attributed to the inhibition of DNA repair mechanisms, we performed an additional experiment in which we exposed mice to a single round of radio-thermotherapy in the presence or absence of ganetespib and evaluated the kinetics of foci formation by P53-binding protein 1 (53BP1), an important mediator in orchestrating the DNA damage response and repair pathway decision making [84,85]. We observed that the addition of ganetespib to radio-thermotherapy did not influence the amount of 53BP1 foci per cell, arguing that double-strand break repair is similarly affected at these timepoints in vivo. We concluded that ganetespib can enhance the effects of radio-thermotherapy in vivo, although the response was not uniform.

## 4. Conclusions and Future Perspectives

Cancer treatments using hyperthermia have been applied to a spectrum of tumor types. Their efficacy is limited, however, by the activation of the HSR. In recent years, several well-tolerated inhibitors of HSP90 have been developed and, some of them, tested in clinical settings. The goal of this study was to evaluate ganetespib, one of the most promising HSP90 inhibitors, as a potential sensitizer in therapies combining hyperthermia with radio- or cisplatin-based chemotherapy.

As inhibition of HSP90 and other chaperones induces a compensatory HSR activation [76,77], we first investigated the effects of various ganetespib pre-exposure durations on the treatment outcomes in vitro. Given the long list of HSP90 client proteins, a longer pre-exposure might increase thermo-sensitization by inducing a systemic proteome malfunction, which could counteract the compensatory HSR. Our results show, however, at least in vitro, that the strength of HSR, manifested by overexpression of HSP70 (Figure 1D,E), overwhelms any potential benefits of long HSP90 pre-exposure, underscoring the redundancy in the cellular chaperone system. We conclude, therefore, that further experiments should adopt the short preincubation schedule, and that this approach would likely be superior in clinical settings. Alternatively, a combination of inhibitors targeting different HSPs (e.g., HSP70 and HSP90) could be considered [86], but would be more likely to induce adverse systemic effects and complicate translation into clinical settings.

Using the optimized treatment conditions, we then used advanced image analysis pipelines based on the cell cycle reporter Fucci2a, to confirm our earlier findings [56] that ganetespib inhibits DNA repair, induces apoptosis and activates cell cycle checkpoints. We also applied clonogenic survival assays to confirm that treatment is effective in a panel of 10 different cancer cell lines (Figure 2D), and that ganetespib can reduce the thermal dose and temperature required for sensitization (Figure 3C,D). These results are in line with multiple earlier studies relying on various inhibitors, treatment schedules and experimental models [44,87,88,89,90,91,92,93]. It is thus clear that, at least in vitro, HSR disruption is an effective sensitization strategy.

To investigate the mechanisms driving these sensitization effects in an unbiased manner, we turned to SILAC-based proteomics. Interestingly, we found that the total number of proteins detected at 24 h after hyperthermia (~2000) was only half of the number detected under all other conditions (~4000), and relatively stable between replicates, which may have resulted in underestimation of the number of differentially abundant proteins. This reduction could, potentially, be due to changes (e.g., denaturation and aggregation, [94,95]) causing the proteins to precipitate from solution during sample preparation, or due to post-translational modifications (e.g., ubiquitination), leading to lower detection probability. Despite the differences in the total number of detected proteins, we found that combined treatment with ganetespib and hyperthermia had effects on a number of different pathways, indeed including DNA repair. Although we could not validate the altered expression of several DNA repair proteins (Figure 4), we did observe a functional impairment of DNA repair in our in vitro experiments (Figure 5). In addition, our results are well aligned with the documented effects of hyperthermia on DNA repair [7] and the relevance of HSP90 therein [80,81,96,97,98,99]. Apart from the DNA damage response, our data suggest that genes regulated by MYC are uniquely differentially expressed after combined exposure to ganetespib and hyperthermia, which is explained by studies showing that MYC is a client of HSP90 [79]. The transcription factor MYC is an important player in many processes, including cell cycle progression, metabolism, chromosomal instability, DNA damage repair, and differentiation [100]. Given that our intention was to explore the global proteome changes after the different combination treatments, rather than focus on individual proteins, we decided not to pursue validation of additional proteins that had altered abundance in our mass-spectrometry data.

Finally, we evaluated the efficacy of the hyperthermia and ganetespib combination as a sensitizer to cisplatin or radiation in a murine model of cervix cancer. To our knowledge, this is the first study to examine HSP90 inhibitors as hyperthermia sensitizers in combination with chemo- and radiotherapy in vivo. We adopted this approach because adding ganetespib—a well-tested and generally safe drug [49,71,72,101]—to the already-applied regimens of hyperthermia and radio/chemotherapy may be the path of least resistance for clinical adoption. Somewhat surprisingly, we found that neither the single-agent treatments, nor the various combinations of two agents, including hyperthermia with radiation or cisplatin, significantly affected tumor growth. Likewise, we found no effect of double combination treatments that included ganetespib. In our previous study, using a syngeneic rat model, some of these combinations did reduce tumor outgrowth [96], and similar effects were reported by others [42,44,90]. It appears, therefore, that the tumors of the mouse model used in the current study are relatively resistant to the applied doses of radiation and cisplatin. We also observed no significant effects in the triple-agent arm including cisplatin. One factor that likely caused or contributed to this outcome was the twofold reduction of cisplatin and ganetespib dosage (as compared to the radiation arm) due to toxicity issues encountered in pilot experiments (not shown). We did detect a considerable delay in tumor outgrowth in the triple-agent arm including irradiation. Interestingly, while the tumor growth in two out of six animals in this group was similar to other groups, it was significantly delayed in the remaining four animals, with one of them showing no growth at all (Figure 6). As our in vitro results suggested that an altered DNA repair capacity at least partially drives the enhanced efficacy of the combination therapies, we evaluated DNA repair capacity in vivo by quantifying 53BP1 foci kinetics. Surprisingly, however, ganetespib did not appear to affect DNA repair capacity. Although this discrepancy might be explained using a different marker, inadequate timepoints, or the presence of non-responding animals (Figure 6C), it can also be hypothesized that this unaltered DNA repair capacity explains the mixed outcomes in vivo and should play part in follow-up studies. Furthermore, as our in vitro findings demonstrate that pre-activation of the HSR can impede treatment efficacy, it is worthwhile to evaluate the extent of its activation in tumor biopsies, especially since cancer cells misuse components of the HSR for their survival, which correlates with tumor progression and metastatic potential [24,46,47,102], and this might be more pronounced in vivo. Consequently, insights into the workings of the HSR and DNA repair in vivo might shed light on the discrepancy between the in vitro and in vivo situation.

What is the future of ganetespib, or other HSP90 inhibitors, for the improvement of hyperthermia-based cancer treatments? The limitations of our study notwithstanding, the addition of ganetespib improves the effectiveness of heat treatment in vitro and seems beneficial in vivo when applied in combination with radio-thermotherapy. Although many studies successfully sensitized cancer cells to hyperthermia using other HSR inhibitors [34,35,36,37,38,39,40,41,42,43,44,45], together with our proof-of-concept paper [56], the clinical compatibility of ganetespib makes this study directly clinically relevant. We believe, therefore, that testing this novel treatment combination in a clinical setting would be a valuable addition to the current clinical trial landscape—of the approximately 75 clinical trials on ClinicalTrials.gov (accessed on 12 September 2022) that test HSP90 inhibitors, the number that include radiotherapy is low (ganetespib: NCT02389751, NCT01554969; onalespib: NCT02381535; AUY-922: none). Additional studies that evaluate the potential of HSP90 inhibition for the hyperthermia-based treatment of other tumor types should also be considered. In parallel, future insights in the workings of the HSR and effective measures to inhibit its components will be of great interest, as it is known that translational power is lacking for chaperone inhibitors, including those for HSP90 [103]. Efforts should also be made to test the thermo-sensitizing effects of other HSP90 inhibitors, because these inhibitors can bind different regions of HSP90 [104], potentially affecting different functions of HSP90. Combination therapies that inhibit all HSP90 paralogs [105] can also be considered. Finally, various in vitro studies have shown that an increased dependency on client proteins, including EGFR, Bcr-abl and c-Met enhances the sensitivity to HSP90 inhibitors, and similar trends have been spotted in clinical trials [103,106]. It is therefore worth evaluating how HSP90 client dependency affects the efficacy of hyperthermia-based treatments. This will also help to consider ganetespib for the hyperthermia-based treatment of various tumor types, as well as patient selection.

## Figures and Tables

**Figure 1 cancers-14-05250-f001:**
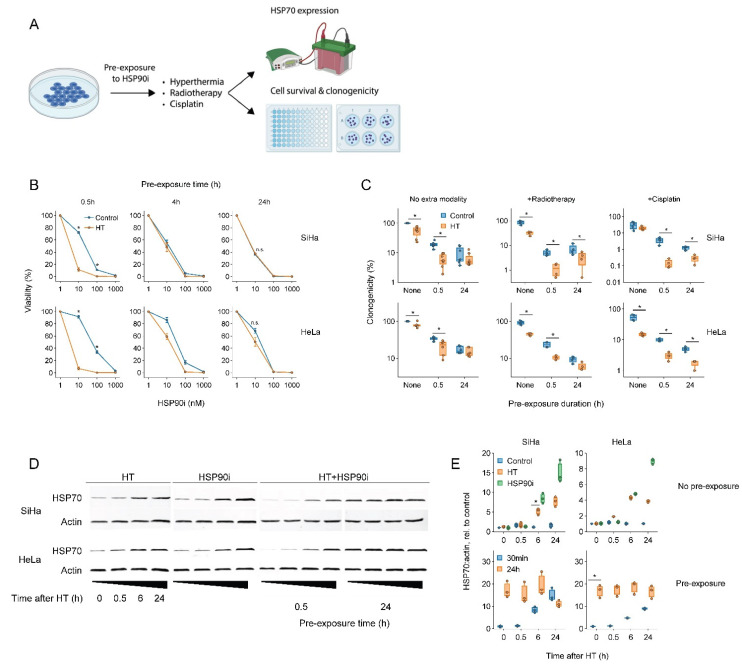
Short pre-exposure to ganetespib maximizes the radio- and chemo-sensitization by hyperthermia. (**A**) Experimental setup. Cervix cancer cells were pre-exposed to ganetespib (HSP90i) for different durations and treated with different combinations of hyperthermia (HT), 3.3 μM of cisplatin and 2 Gy of radiotherapy. (**B**) Cell viability 72 h after treatment with HT and HSP90i. Noteworthy significant differences are marked with an asterisk. Not significant: “n.s.”. (**C**) Effect of pre-exposure with 30 nM of HSP90i on the clonogenicity after radio- and chemo-thermotherapy. (**D**) Immuno-blot showing levels of HSP70 and beta-actin (loading control) at various time points after HT and HSP90i. The uncropped blots are shown in Appendix A. (**E**) quantification of (**D**).

**Figure 2 cancers-14-05250-f002:**
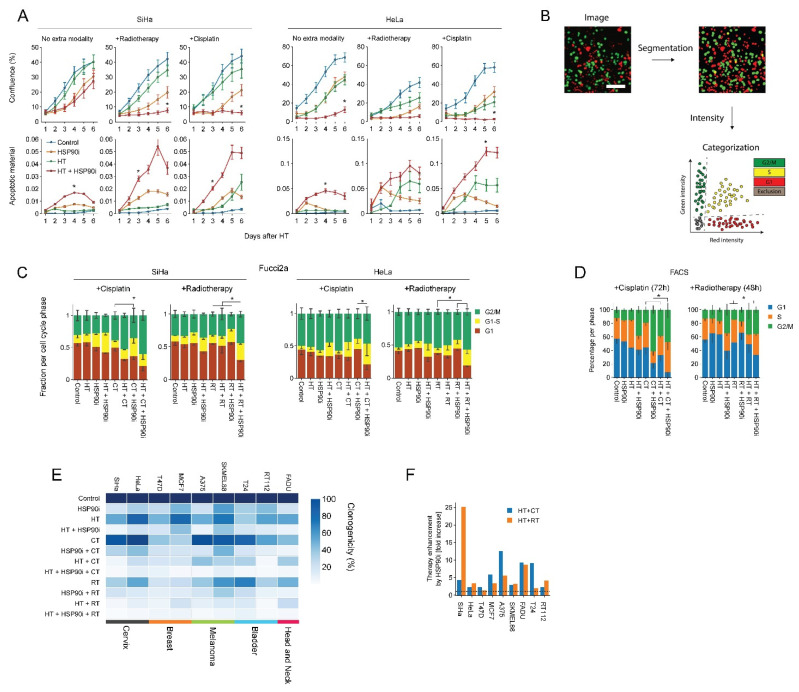
Ganetespib improves the efficacy of hyperthermia-based treatments in vitro. Cervix cancer cells were exposed to combinations of hyperthermia (HT), 30 nM of ganetespib (HSP90i), 2 Gy of radiotherapy (RT), and 3.3 μM of cisplatin (CT). (**A**) Cell growth and apoptosis were monitored and quantified over the course of six days with the IncuCyte S3 imaging platform. Noteworthy significant differences are marked with an asterisk. (**B**) Cell cycle analysis pipeline after radio- and chemo-thermotherapy using Fucci2a-transfected cervix cancer cells. Cells were imaged daily after treatment, and image segmentation was performed, followed by quantification of fluorescence intensities and thresholding to detect cells in the different phases of the cell cycle. Scale bar: 100 μm (**C**) Quantification of cell cycle distribution after radio- and chemo-thermotherapy 48 and 72 h after HT, respectively, based on the analysis of cells transduced with the Fucci2a reporter. (**D**) FACS-based cell cycle analysis at identical time points. Noteworthy significant differences between the percentage of cells in G1 are marked with an asterisk. (**E**) Heatmap showing clonogenicity after radio- and chemo-thermotherapy of a cancer cell line panel. (**F**) Enhancement ratio of chemo- and radio-thermotherapy outcomes in the presence of ganetespib, displayed as a fold-increase relative to the respective treatment without ganetespib. The horizontal dotted line marks no enhancement.

**Figure 3 cancers-14-05250-f003:**
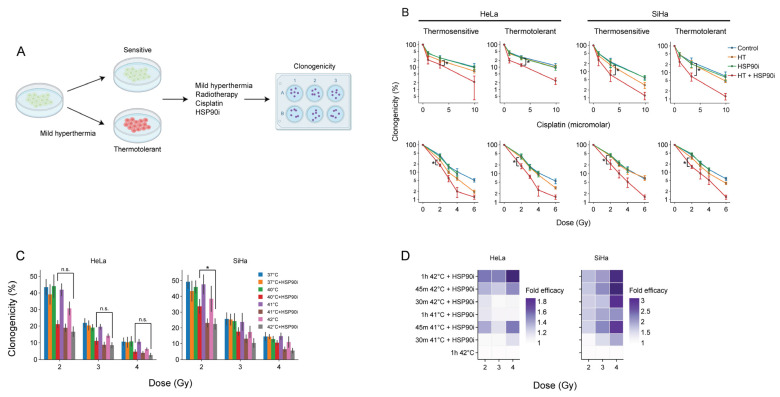
Ganetespib re-sensitizes thermotolerant cells and promotes the effects of hyperthermia-based treatments at lower thermal doses. (**A**) Experimental setup. Cervix cancer cells were rendered thermotolerant by hyperthermia (HT) treatment 24 h before exposure to 2 Gy of radiotherapy or 3.3 μM of cisplatin in the absence or presence of 30 nM of ganetespib (HSP90i). Clonogenicity was used as a readout for therapy efficacy. (**B**) Clonogenicity of control and thermotolerant cells after exposure to various chemo- and radio-thermotherapy regimens. Noteworthy significant differences are marked with an asterisk. Not significant: “n.s.”. (**C**) Clonogenicity after treatment with radio-thermotherapy at lower thermal doses, in the presence or absence of ganetespib. (**D**) Enhancement ratio of chemo- and radio-thermotherapy outcomes at lowered thermal dose in the presence of ganetespib, displayed as a fold-increase relative to the respective treatment without ganetespib. The horizontal dotted line marks no enhancement.

**Figure 4 cancers-14-05250-f004:**
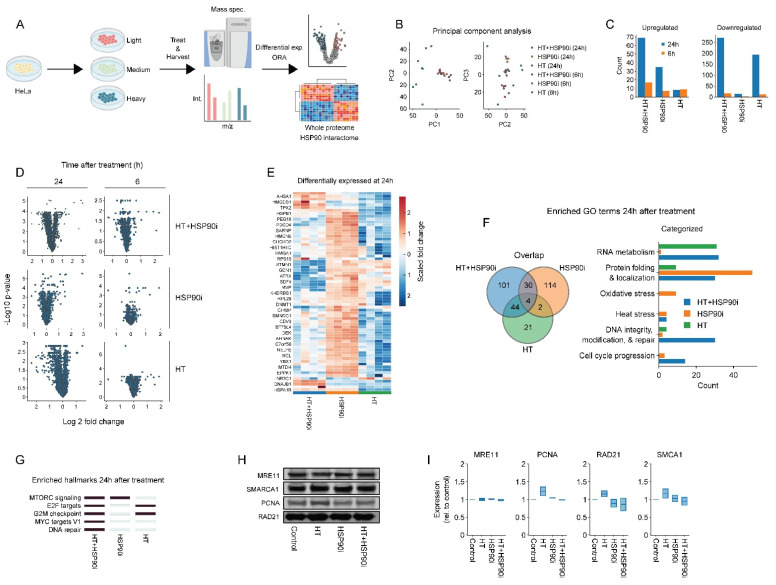
Proteome alterations caused by ganetespib and hyperthermia. (**A**) Experimental setup. The HeLa cells were labeled in SILAC media, exposed to combinations of 30 nM of ganetespib (HSP90i) and hyperthermia (HT), and processed for proteomics and downstream analysis. (**B**) Principal component analysis. (**C**) Number of up- and downregulated proteins (log2 fold change higher than 0.5 or lower than −0.5). (**D**) Volcano plots of all conditions. (**E**) Heatmap of proteins found to be up- or downregulated under at least one condition. Overrepresentation analysis on up- and downregulated proteins yielded results presented in panels (**F**,**G**). (**F**) Venn diagram and hand-curated categories of gene ontology (GO) terms found to be enriched per condition. (**G**) Heatmap displaying enriched hallmark gene sets (marked in dark blue). (**H**,**I**) Immuno-blots and quantifications showing the expression of proteins of interest 24 h after HT. Equal protein loading was ensured by a Lowry assay. The uncropped blots are shown in Appendix A.

**Figure 5 cancers-14-05250-f005:**
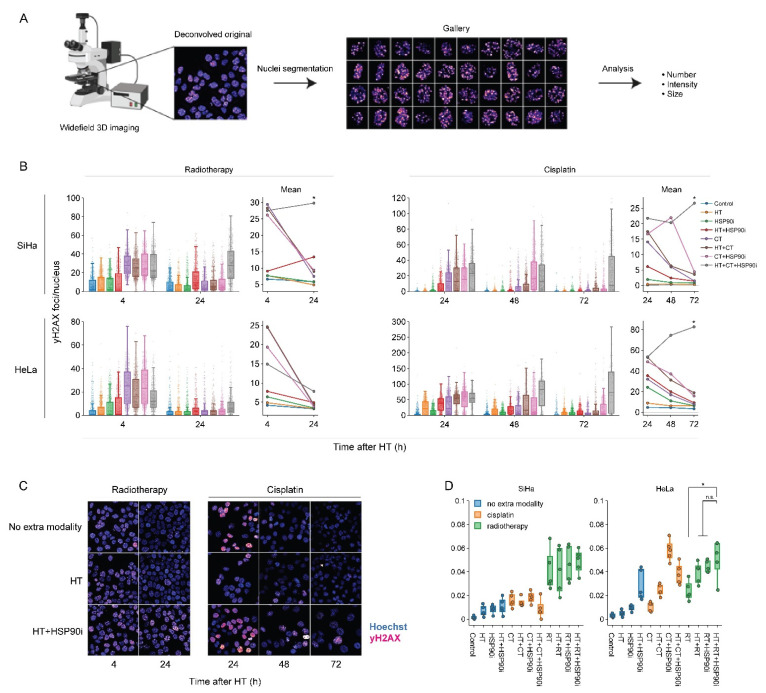
Ganetespib hampers DNA double-strand break repair after radio- and chemo-thermotherapy. (**A**) Image processing pipeline. Cells were imaged in 3D mode and images were deconvolved. Maximum intensity projections were then generated and used as input for the segmentation of cell nuclei, micronuclei and yH2AX foci. (**B**) Boxplots showing the number of yH2AX foci per cell, and line plots displaying the mean number of yH2AX foci after combinations of 2 Gy of radiotherapy (RT), 3.3 μM of cisplatin (CT), and hyperthermia (HT) in the presence or absence of 30 nM of ganetespib (HSP90i). Noteworthy significant differences are marked with an asterisk. Not significant: “n.s.”. (**C**) Representative images used for quantification. Scale bar: 30 μM. (**D**) Percentage of cells harboring at least one micronucleus 48 h after HT.

**Figure 6 cancers-14-05250-f006:**
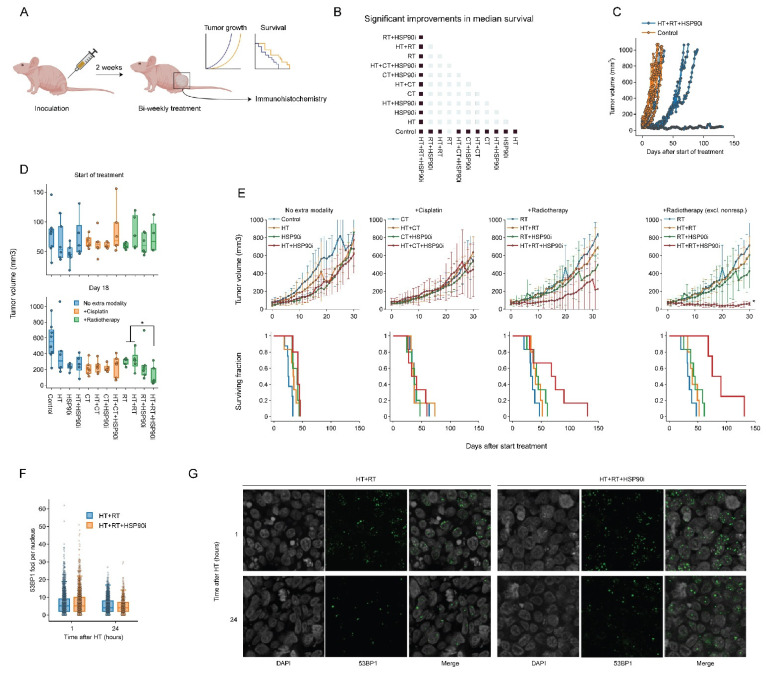
Ganetespib improves radio-thermotherapy in a cervix cancer mouse model. (**A**) Experimental setup. Hind paws of nude mice were inoculated with SiHa cells and animals were followed until the resulting tumors reached approximately 50 mm^3^. The animals were then treated with combinations of hyperthermia (HT), radiotherapy (RT), cisplatin (CT), and ganetespib (HSP90i) twice a week. Tumor growth was monitored, and mice were sacrificed when the volume reached 1000 mm^3^. A subgroup of mice was treated once and sacrificed to allow monitoring of DNA double-strand break induction and repair. (**B**) Heatmap showing significant (fold) improvements (marked in dark blue) in median survival. (**C**) Tumor volume of untreated cohort versus the cohort treated with HT+RT+HSP90i, showing non-responders in the latter. (**D**) Tumor volume per condition at the start of the treatment schedule and on day 18. (**E**) Tumor volume (top) and Kaplan–Meier analysis (bottom). For the RT arm, results are shown with inclusion and exclusion of non-responders. Noteworthy significant differences are marked with an asterisk. (**F**) Boxplot showing the number of 53BP1 foci per cell 1 h and 24 h after treatment. (**G**) Representative images used for the quantification shown in F. Scale bar: 10 μM.

## Data Availability

The data presented in this study are openly available in interactive form via figlinq.com (https://create.figlinq.com/dashboard/e.m.scutigliani:858 (accessed on 18 October 2022)).

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
