# Peer review of "Evaluation of the Heat Shock Protein 90 Inhibitor Ganetespib as a Sensitizer to Hyperthermia-Based Cancer Treatments"

_cancers, 2022, doi:10.3390/cancers14215250_

Round 1
Reviewer 1 Report
In this manuscript, the authors have determined whether HSP90 inhibitor, ganetespib, boosted the effect of radio- and chemotherapy when combined with hyperthermia. They firstly optimized the incubation time and dosage of ganetespib to maximize its potential and then discovered that ganetespib sensitized cells to radiotherapy or cisplatin among various tumor types. Next, They found that ganetespib re-sensitized thermotolerant cells. Furthermore, they explored the changes of proteomes on HeLa cells treated with ganetespib and/or hyperthermia to identify the underlying mechanism. The results suggested that significant changes were most significantly involved in DNA repair and cell cycle progression. Next, they confirmed that DNA repair pathways were further inhibited with combined treatment. Lastly, they validated that effect of ganetespib in vivo and discovered that ganetespib enhanced the effect when hyperthermia was combined with a radio-thermotherapy, not cisplatin treatment. Overall, they suggested that HSP90 inhibitors could serve as a potential sensitizer for hyperthermia-based cancer treatments. However, there are some remaining questions to be answered:
1, Could the authors elaborate the Fig. 1D in more details? There are four bands in each group, are they representing 4 time points or replicates?
2, Have the authors considered other assays to validate the proteome results since expression levels of 4 DNA repair proteins are not significantly decreased. For example, qPCR to measure the mRNA level? Cell fractionation assay to examine the localization? Functional assay to determine the protein activities?
3, It seems like the authors performed separated in vivo experiments to evaluate DNA damage response. Have the authors performed IHC analysis on groups in Fig. 6E instead? Since ganetespib to radiotherapy worked for 4 of 6 mice, it might be promising to evaluated the tumor samples from this group? Does combined treatment exhibited more cell death (cleavage caspase), more DNA damage in these tumor slides?
4, Does other chemotherapeutic drug outperform cisplatin in treatment? Have the authors considered to try other chemotherapy agent instead?
Author Response
Dear reviewer,
We would like to express our gratitude for reviewing our manuscript. The reviewing process has provided us with useful input, and has significantly improved the quality of our manuscript.
Below you will find a point-by-point response to your comments, and we are looking forward to the next step in the submission procedure.
Thank you, on behalf of all authors,
Enzo Scutigliani.
- Could the authors elaborate the Fig. 1D in more details? There are four bands in each group, are they representing 4 time points or replicates?
Thank you for critically checking the figures. We made adjustments to the figure, and specified in the legend that this immunoblot displays multiple time points.
- Have the authors considered other assays to validate the proteome results since expression levels of 4 DNA repair proteins are not significantly decreased. For example, qPCR to measure the mRNA level? Cell fractionation assay to examine the localization? Functional assay to determine the protein activities?
We acknowledge that this statement deserves extra attention in the manuscript, and we therefore included an extra sentence in the discussion section that clarifies our thinking process. In our opinion, of all options to validate proteomics results, checking protein expression by immunoblotting can be regarded as the most comparable to what is measured by proteomics. As our goal was to specifically validate the proteomics-related results, we did not consider to evaluate mRNA levels and cellular localization, although we do agree with you that these can provide important insights into what happens to these proteins when cells are treated with hyperthermia and HSP90 inhibition. We considered a functional assay to look at protein activity, but since we could not validate the proteomics results, we decided to undertake experiments that look at the functional outcome of DNA repair in a broader sense, and these are the experiments in which we quantified gamma-H2AX foci and micronuclei occurrence after clinically relevant combination therapies (Figure 5).
- It seems like the authors performed separated in vivo experiments to evaluate DNA damage response. Have the authors performed IHC analysis on groups in Fig. 6E instead? Since ganetespib to radiotherapy worked for 4 of 6 mice, it might be promising to evaluated the tumor samples from this group? Does combined treatment exhibited more cell death (cleavage caspase), more DNA damage in these tumor slides?
This is an excellent point and we agree that it would be very useful to evaluate whether there are differences in the ability to repair DNA between non-responders and responding animals. As the animals assigned to the survival study could not be used for tumor biopsies until the tumor reached a certain size, and since these mice live for a relatively long period after undergoing the treatment regimen, we reasoned that performing additional experimentation on these tumors would result in a highly biased view on DNA repair capacity. We therefore decided to conduct a separate experiment to have the ability to harvest the tumors after a single round of therapy at a set time point that is similar for all relevant conditions. The disadvantage of this setup is of course that we cannot evaluate which animals are responding to therapy. We discuss the limitations of our setup and potential follow-up studies in the discussion section and to further clarify to the reader that the survival outcome and IHC analysis originate from different experiments, we made minor adjustments to the results section.
- Does other chemotherapeutic drug outperform cisplatin in treatment? Have the authors considered to try other chemotherapy agent instead?
If we understand your question correctly, you’re asking whether we considered testing other drugs in vitro and in vivo in combination with hyperthermia and HSP90 inhibition. Since our study, especially the in vivo part, revolves around cervix cancer, we tested modalities that are most frequently administered in combination with hyperthermia for this indication. We agree that it would be very interesting to expand our studies to other tumor types and evaluate how HSP90 inhibition and hyperthermia can contribute to the treatment of other patient groups, and we made textual adjustments in the discussion section to bring this point to the attention of the reader.
Reviewer 2 Report
In present article, authors have evaluated if HSP90i can sensitize Chemo-theromotherapy and Radio-thermotherapy in cervical cancer cell models and murine models. Authors have done tremendous work and asked important questions. This article will be very much useful for optimizing and developing new therapies for Cervical cancer. Although, article is well written some items for improvement are listed below:
1: Figure 1c: Why is 0.5 pre exposure time + Radiotherapy or Cisplatin so significance was shown? I believe they are significant. What was cisplatin and radiation treatment concentration used in this experiment?
2: Figure 1d: Authors should label properly including time after HT.
3: Authors should discuss results better. Especially for figures 1 and 2. What phase of cell cycle are cells arrested at? From results in SiHa cells I see G2/M cell cycle arrest but HeLa cells not so obvious. It would be best to write it in text. Did authors perform statistical analysis for this, was it significant?
Author Response
Dear reviewer,
We would like to express our gratitude for reviewing our manuscript. The reviewing process has provided us with useful input, and has significantly improved the quality of our manuscript.
Below you will find a point-by-point response to your comments, and we are looking forward to the next step in the submission procedure.
Thank you, on behalf of all authors,
Enzo Scutigliani.
- Figure 1c: Why is 0.5 pre exposure time + Radiotherapy or Cisplatin so significance was shown? I believe they are significant. What was cisplatin and radiation treatment concentration used in this experiment?
This is a good point, we changed the annotation of the figure and marked all significant differences. Thank you for pointing out that a documentation on concentrations/dosages was missing, we made sure that detailed information on this is now given in figure legends.
- Figure 1d: Authors should label properly including time after HT.
Thank you for critically checking the figures. We made adjustments to the figure, and specified in the legend that this immunoblot displays multiple time points.
- Authors should discuss results better. Especially for figures 1 and 2. What phase of cell cycle are cells arrested at? From results in SiHa cells I see G2/M cell cycle arrest but HeLa cells not so obvious. It would be best to write it in text. Did authors perform statistical analysis for this, was it significant?
We agree that the results with regards to the cell cycle analyses are underrepresented in the manuscript, and made minor textual adjustments to elaborate on these experiments without deviating from the main message that we wish to convey. Statistical analysis has also been implemented in the figure and the legend, and we thank you for providing suggestions to make this figure more informative.